# Stephan Oroszlan and the Proteolytic Processing of Retroviral Proteins: Following A Pro

**DOI:** 10.3390/v13112218

**Published:** 2021-11-04

**Authors:** Ronald Swanstrom, Wesley I. Sundquist

**Affiliations:** 1Department of Biochemistry and Biophysics, University of North Carolina at Chapel Hill, Chapel Hill, NC 27599-7295, USA; 2Department of Biochemistry, University of Utah School of Medicine, Salt Lake City, UT 84112-5650, USA

**Keywords:** Oroszlan, retroviruses, HIV-1, protease, capsid

## Abstract

Steve Oroszlan determined the sequences at the ends of virion proteins for a number of different retroviruses. This work led to the insight that the amino-terminal amino acid of the mature viral CA protein is always proline. In this remembrance, we review Steve’s work that led to this insight and show how that insight was a necessary precursor to the work we have done in the subsequent years exploring the cleavage rate determinants of viral protease processing sites and the multiple roles the amino-terminal proline of CA plays after protease cleavage liberates it from its position in a protease processing site.

## 1. Creating a Framework

We all know our colleagues on multiple levels. We meet them at meetings. We read at least some of their papers. We can be influenced by their work to either add new meaning to our own work or even to guide our work in new directions. There is also the poignant exercise of viewing the entire body of work of a lost colleague through the lens of PubMed to refresh memories but also to learn unappreciated details that go with the life of a productive scientist. Steve Oroszlan was by any measure a productive and successful scientist who left an impact on the field of retrovirology. In this short remembrance, we will review highlights of his career that led to the characterization of the ends of retroviral virion proteins and thus defined protease cleavage sites. Knowledge of the cleavage sites was essential information for understanding viral Gag protein processing by the viral protease (PR) and virion maturation. Here, we will place Steve’s work as the foundation for our later studies and thereby try to thank him for his contributions to these important questions. Specifically for this remembrance, Steve’s identification of the amino acid proline as the amino terminus of a diverse set of retroviral CA/capsid proteins deepened our understanding of the regulation of the rate of cleavage at different protease processing sites, and to the critical role this amino acid plays in CA conformation, virion maturation and capsid function. Our goal is not to provide a comprehensive review of all of Steve’s work but rather to place it in the context of how it influenced our own work. More general reviews of the topics that we touch upon will quickly reveal the many important contributions by other colleagues that, with apologies, are not included in this remembrance. We are writing this perspective in the first-person plural even though this is actually our first collaboration; the following use of “we” will most often reflect work done in our individual labs, where we have each had the good fortune to work with many talented scientists.

Reviewing Steve’s publications indicates the path he followed to the point where we became aware of his work. We can be excited for a young Stephan Oroszlan publishing a Nature paper in 1964 on the use of polyglucose in the purification of viruses using density centrifugation to overcome the problem of the high osmolarity of sucrose [1]. By the next year, Steve had moved into the then RNA tumor virus field by applying his purification approach to the murine leukemia virus [2]. This early interest in purification was an essential part of his portfolio that would allow the subsequent sequencing of virion proteins (more later). Another feature of Steve’s work is that he explored parallel questions in different retroviruses and thereby contributed substantially to the powerful concept of comparative virology. This can be seen in papers examining a reptilian retrovirus [3], murine leukemia virus [4], a hamster retrovirus [5], several cat viruses [6,7], a baboon virus [8], woolly monkey and gibbon ape viruses [9], bovine leukemia virus [10], avian myeloblastosis virus [11], endogenous retroviruses [12,13,14], HTLV-1 [15], Mason-Pfizer monkey virus [16], equine infectious anemia virus [17], mouse mammary tumor virus [18], two simian immunodeficiency viruses (SIVmne and SIVmac) [19], HIV-1 [20], and probably others.

The study of so many different retroviruses (often through productive collaborations) had many benefits, including two we will mention here. First, his characterization of virion proteins included their immunological features and, specifically, the observations that the internal virion structural proteins could be used to describe families or groups of viruses with cross-reactive antibody specificity. This observation provided the origin of the name of the viral *gag* gene and its associated Gag protein, standing for group-specific antigen [4,5,6,7,21,22]. Steve was so central in defining both the virion proteins and virus relationships through these immunologic cross-reactivities that he was included in ad hoc committees for both the standardized naming of the retroviral proteins [23] (Figure 1a) and the final naming of the then-new AIDS virus as human immunodeficiency virus type 1 (HIV-1) [24,25]. A second lasting contribution from his work with these diverse viruses was the ability to find commonality conserved across their great biological differences, which, of course, helps to identify critically important functions.

It is also appropriate to pay homage to the special skill of protein sequencing that Steve brought to the retrovirus field, often together with his longtime collaborators Lou Henderson, Terry Copeland, and Alan Schultz. It would be easy to underestimate this contribution, given that protein sequences are now inferred based on gene sequences. However, proteins (and RNAs) are the functional units of virology and defining the ends of the mature virion proteins that started life out as part of large precursor proteins revealed the cleavage sites of the viral protease (see below). In addition, the inability to sequence the end of the Gag protein led to the discovery that the Gag precursor and its N-terminal product, the MA/matrix protein, are myristoylated [26,27]. This observation led to the understanding that the N-terminal MA protein of Gag is responsible for targeting both the Gag and Gag-Pro-Pol precursors to the cellular plasma membrane to initiate virion formation.

## 2. Defining Protease Cleavage Sites

The large-scale growth and purification of a diverse set of retroviruses to allow protein sequencing of their virion proteins was largely accomplished solely by Steve Oroszlan and his group. This work started when Steve was working with Ray Gilden and continued, transitioning to Steve as the leader for over 15 years [7,8,9,10,11,12,13,14,15,17,18,19,28,29,30,31,32,33,34,35]. Their work, together with the demonstration that the virion proteins are synthesized as a larger precursor [36,37], and the detection of a viral protease in virions [38,39], provided the framework for the entire field of retroviral assembly and maturation, which is still very active. As a result of Steve’s work, we now know that Gag and Gag-Pro-Pol precursor proteins contain between five and ten viral protease cleavage sites (depending on the retrovirus).

Studies with peptide substrates have shown that the viral protease binds to a seven amino acid stretch of protein. The amino acid positions around the protease cleavage site are labeled following the convention: P4 P3 P2 P1/P1’ P2’ P3’, where the/indicates the cleavage site position (also called the scissile bond). These position designations hold for any cleavage site, regardless of the protein or local amino acid sequence, allowing equivalent positions to be compared between different cleavage site sequences. We will try to be careful in this presentation since we will also be talking about the proline that becomes the N-terminal amino acid of the mature CA/capsid protein, creating the confusing situation where the P1’ amino acid of the cleavage site that creates the N terminus of CA also creates Pro1 as the amino acid label of the first amino acid of mature CA.

Rather than detail the many studies of viral cleavage sites, we present two important concepts that emerged from these analyses. The first concept is that the retroviral cleavage sites are remarkably heterogeneous. This heterogeneity might not be surprising across different retroviruses because their proteases could have evolved different specificities, and this has happened to some extent. The more important point, however, is that the heterogeneity among the cleavage site sequences of even a *single* set of retroviral Gag and Gag-Pro-Pol precursor proteins is also dramatic to the point of confusion (at least for divining the basis of protease specificity). This point is evident in the viral protease cleavage sites for HIV-1. As can be seen in Figure 1b, there is great sequence diversity across these different sites. At first glance, about all that is common is that the P1 and P1’ amino acids are hydrophobic amino acids and they are devoid of beta-branched amino acids (Ile and Val). Beyond that, very little “conservation” is evident. Moreover, it is unusual for most proteases to cleave at sites that have a P1’ proline, yet several such sites exist among the HIV-1 cleavage sites. There is good evidence that these sites are cleaved at different rates, especially in vitro (Figure 1c). However, it can be difficult to parse the contributions of suboptimal cleavage site sequences versus unfavorable structural contexts within the full precursor protein. For now, we will consider just the sequence effects and not the context. In general terms, the cleavage at the N terminus of the NC/nucleocapsid protein occurs first to allow condensation of NC with viral RNA, followed by cleavage at the N terminus of CA to release CA from the membrane-bound MA. This likely liberates CA to start to reassemble around the condensed NC/RNA complex (Figure 1d).

While any comparison of cleavage sites for single retroviruses inevitably shows great sequence heterogeneity (albeit with hydrophobic P1 and P1’ amino acids), the unusual mixing of proline, as the only amino acid with a constrained peptide bond, and other hydrophobic amino acids at P1’ in protease cleavage sites led several to suggest that perhaps cleavage sites could be divided into at least two groups based on the identity of the P1’ amino acid [40,41,42,43]. Notably, the P1’ amino acid that defines the new N terminus of CA after cleavage is always a proline. This conservation can be seen in a partial compilation of cleavage sites from different retroviruses that create the CA N-terminus, as shown in Table 1. Nature rarely gives such easily observable clues, but this one was only evident after Steve Oroszlan had generated enough N-terminal sequences of virion proteins from different retroviruses to allow such a comparison. At the time, it was clear that this was an important observation, although further structural and functional studies would be required to reveal multiple roles performed by CA Pro1 following release from the Gag polyprotein precursor. In recognition of his foundational contribution, we will refer to this important amino acid as the Oroszlan Pro1.

## 3. The Oroszlan Pro1 Amino Acid Regulates the Structure and Function of the Viral Capsid

By definition, a protease cleavage site must be accessible to the protease. Thus, on first principles, we can anticipate that cleavage sites will exist in minimally-structured loop or linker regions to allow viral protease accessibility, particularly as the cleavage site must adopt an extended beta-sheet conformation within the active site of PR. This rule is broken by the slowly cleaved HIV-1 CA/SP1 cleavage site, which is an important exception because the requirement for the unfolding of the helical CA/SP1 site accounts for the slow rate of cleavage at this site and also for the activity of the betulinic acid maturation inhibitors, which stabilize the helical assembly, and thereby, retard cleavage rates even further [44,45,46,47,48,49]. Unlike this exceptional case, the cleavage site immediately upstream of CA starts out largely unstructured in the immature virion [50,51,52] (Figure 2a), even though CA itself plays a central role in organizing the immature viral capsid [51]. Gag processing then drives the formation of the mature viral capsid. This story is most easily told using HIV-1 as the example since we have the most structural information for this virus, although the key elements appear to be conserved across orthoretroviruses. What we see is that cleavage at the MA/CA junction releases CA from being tethered to the viral membrane and induces conformational changes at the newly generated CA N-terminus that are mediated by the newly released Pro1 residue. The story of how we got to our current understanding of the role of Steve Oroszlan’s Pro1 residue is an interesting one, and it nicely illustrates how science progresses as different groups add new insights and apply emerging technologies to gain a greater understanding of what has turned out to be remarkably complex and intriguing biology.

When we first determined structures of the mature N-terminal domain of the HIV-1 CA protein with Summers, Hill, and colleagues [53,54], we immediately noticed that the CA Pro1 residue was buried back into the body of the protein (Figure 2). This is possible because the first N-terminal 13 residues of the newly processed CA protein adopt a β-hairpin structure in which CA residues 9–13 form a β-strand that extrudes away from the body of the domain, residues 6–8 form a loop that reverses the chain direction, and residues 1–5 form a second, pairing β-strand that projects back into the domain, ending in the Pro1, which is buried into a well-defined hydrophobic pocket within CA (Figure 2b). When the amide bond at the MA-CA junction is cleaved, the Pro1 pyrrolidine ring nitrogen becomes a secondary amine, with a pKa of ~9, which means that it will become protonated and carry a positive charge under physiological conditions. This protonated CA Pro1 residue forms a salt bridge with the buried side-chain carboxylate of CA Asp51, and this interaction occupies a central position within an even more extensive hydrogen-bonding network (Figure 2b and Figure 3). Thus, MA-CA proteolysis induces the CA N-terminus to convert from an unstructured polypeptide that can be accessed by the viral protease into a well-defined β-hairpin conformation (Figure 2a). Analogous proteolysis-induced conformational changes involving the formation of a buried N-terminal salt bridge and new hydrophobic interactions occur in other systems, for example, when inactive serine protease zymogens are converted into active proteases [55]. Thus, salt bridge formation by the new N terminus is a general mechanism by which proteins can change structure and function in response to proteolytic processing. In the case of HIV-1, mutating Pro1 to Leu or Asp51 to Ala abolishes viral infectivity and leads to the formation of aberrant capsids that fail to support reverse transcription when the virus infects a new cell, indicating that the hairpin is important for capsid assembly, reverse transcription, and infectivity [56,57,58,59].

Folding of the CA N-terminal β-hairpin appears to be driven by a series of favorable interactions (and the avoidance of unfavorable interactions), that include: (i) the Pro1-Asp51 salt bridge, (ii) a network of favorable neighboring interactions, including the formation of two additional H-bonds between Pro1 and Gln13 on the pairing strand β-strand (Figure 3b), (iii) β-sheet hydrogen bonding interactions between the two newly formed β-hairpin strands, and (iv) removal of the upstream MA polypeptide, which prevents CA Pro1 burial prior to proteolysis owing to unfavorable steric hindrance. However, none of these favorable interactions really explains the conservation of the Oroszlan Pro1 residue because analogous interactions could, in principle, be made by other amino acids at the CA N terminus. So why is the first CA amino acid always a proline? Three factors seem to dictate the requirement for proline. As a secondary amine, proline has a higher pKa_2_ value than all other (primary) amino acids, and the greater basicity of proline may help to tune the strength of the Pro1-Asp51 salt bridge. More importantly, the aliphatic resides in the Pro1 pyrrolidine make favorable van der Waals contacts with the CA proline binding pocket. Perhaps most importantly, this specialized binding pocket is configured to allow the CA Pro1-Asp51 interaction to function as a “hinge” that accommodates multiple different β-hairpin conformations (see below).

Mature HIV-1 capsids are fullerene cones composed of ~240 CA hexamers and exactly 12 CA pentamers [60,61,62,63,64]. Capsids assemble following Gag proteolysis [65], and the CA subunits make quite different intermolecular contacts in the immature and mature capsid lattices [48,49,51]. In mature CA hexamers (and pentamers), the N-terminal CA helices associate into a ring, and the CA β-hairpins form “crowns” above the central pore of the ring (Figure 3a) [62,63,66,67]. We, therefore, initially hypothesized that β-hairpin formation might serve as a structural switch that helps to drive the conversion of the immature HIV Gag lattice into the mature capsid lattice. Although this may be part of the story, Kräusslich, Briggs, and colleagues have since shown that mature CA–CA subunit interactions can occur in the absence of β-hairpin formation and that proteolysis at the CA/SP1 junction instead serves as the primary structural switch that drives lattice maturation [68]. This is because CA/SP1 cleavage disrupts the six-helix bundle that stabilizes the hexameric Gag building blocks of the immature lattice, thereby promoting mature capsid lattice formation.

We now understand that the β-hairpin actually plays even more interesting roles in helping to stabilize the mature viral capsid and promote viral replication upon infection of a new target cell. Recent studies indicate that the HIV-1 capsid remains largely intact as the virus crosses the cytoplasm and moves into the nucleus [69,70]. Reverse transcription occurs during this time, and we have shown that the intact capsid actually plays an essential role in promoting efficient reverse transcription [71]. However, these observations beg the question of how dNTPs can get into closed capsids to feed reverse transcription. This question was beautifully answered by James and colleagues, who showed that the central pore of the mature CA hexamer can serve as a non-specific channel for dNTP transport [72], with the CA Arg 18 residues forming a basic “collar” that binds the translocating nucleotide triphosphates (Figure 4a). Above this binding site, the β-hairpins can adopt distinct “closed” and “open” configurations that appear to regulate dNTP channel access (Figure 3a). Thus, the β-hairpins appear to function as “gates” that allow dNTPs into the viral capsid (and perhaps also allow the exit of diphosphates and viral rNMPs after RNase H cleavage of reverse-transcribed genomic RNA). β-hairpin gating is pH-dependent because protonation of the CA His12 side chain at the base of the hairpin favors the formation of a second salt bridge between Asp51 and His12 in the “pore open” conformation (Figure 3b). Conversely, when His12 is deprotonated, an intervening water molecule mediates the His12-Asp51 interaction, and this configuration closes the gate. CA Pro1 allows the gate to open and close by functioning as a “hinge” that rotates by more than 35° as the states interconvert (Figure 3b).

A final important function of the CA hexamer pore is to bind the abundant inositol phosphate IP_6_. High levels of IP_6_ are incorporated into virions by binding within the six-helix bundles of Gag hexamers, and these packaged IP_6_ molecules then bind above and below the Arg18 collar of the mature CA hexamer (Figure 4b) [73,74,75,76,77]. The upper IP_6_ molecules bind within a chamber that is surrounded by closed CA β-hairpins. Similar to dNTPs, IP_6_ neutralizes repulsive ionic interactions in the Arg collar and thereby dramatically stabilizes the viral capsid. Thus, the CA β-hairpins function both to gate the nucleotide channel and to help capture the stabilizing IP_6_ molecules, and these different activities are made possible by the intricate network of interactions organized about CA Pro1.

## 4. Turning the P1’ Pro Cleavage Site Question Around

The identification of the unique roles that proline plays as the N-terminal amino acid of the mature CA protein requires that we change the question of why the protease “wants” to use proline as a P1’ amino acid. Rather, the protease must *tolerate* a P1’ proline as a byproduct of its essential role in capsid formation and function. Thus, we can reframe the question of the P1’ proline as: How does the protease accommodate the proline that must be present at the N terminus of mature CA?

A simple comparison of cleavage sites within one retrovirus treats the sites as somehow all equal and assumes that their sequences are recognized equivalently by the protease. Instead, we would like to return to the idea of different rates of cleavage. Using a radiolabeled Gag substrate and purified protease, we observed that in vitro cleavage rates varied by up to 400-fold between the different cleavage sites (Figure 1c) [78]. These rate differences were much less pronounced when the same sequences were tested in a common context (replacing the MA/CA cleavage site), but the rank order was largely conserved [79]. We do not know the absolute rates of cleavage in the context of the maturing virus particle, but the accumulation of different processing intermediates demonstrates that the sites are cleaved at different rates and that the order is similar to that observed in vitro. Many investigators have studied the sequence determinants of protease cleavage sites, but usually in isolation as one site. Recently we took a more global, brute force approach by studying six different cleavage sites in parallel and measuring the relative rates of cleavage of >150 wild type and mutant cleavage sites and their chimeras [79]. The cleavage reactions were carried out with an internal control (the wild-type MA/CA cleavage site sequence) so that all cleavage rates could be scaled relative to the rate of cleavage of this internal control, thereby reducing the effects of assay variation. Moreover, all of the cleavage site sequences were placed within the same context of the MA/CA sequence but with a Gly_3_ buffer on either side of the eight amino acid cleavage site sequence. Thus, this system had elements of a peptide substrate (the eight amino acid cleavage site sequence flanked by the Gly_3_ linker) and elements of the test cleavage sites being in the context of a globular protein (MA/CA).

Having 150+ relative rates of cleavage among diverse sites initially was a daunting data set. However, two insights clarified the sequence determinants that define protease cleavage rates. First, although the mutagenesis was not saturating, no substitutions in either the MA/CA site or the SP1/NC site increased the cleavage rate. This is significant because the SP1/NC site is the first site cleaved, and the MA/CA site is the prototypic cleavage site with a P1’ proline. If we assume that these sites are at or near the optimal sequence for rapid cleavage, then the penalty of having a P1’ proline is apparent in the slower rate of cleavage of MA/CA compared to SP1/NC. Assuming these two sites are at or near the optimal sequence also allowed us to compare how the patterns of cleavage rate changed with amino acid substitutions in the cleavage sites. This led to the second insight.

Figure 5 reproduces our results of the activity of the substitutions tested from P4 to P4’ for the SP1/NC and MA/CA sites [79]. The second insight came from comparing the activity of amino acids specifically in the P2 and P2’. Remarkably, the specificity of the amino acids for these two positions in determining the rate of cleavage by the protease is essentially switched between the two sites. For the Met/Met scissile bond of SP1/NC (P1 hydrophobic/P1’ hydrophobic), the optimal P2 amino acids are valine and isoleucine with aliphatic side chains, and the optimal P2’ amino acids are glutamate and glutamine with charged/polar side chains. Conversely, for the Tyr/Pro scissile bond of MA/CA (P1 hydrophobic/P1’ proline), the optimal amino acid in the P2 position is asparagine with a polar side chain, and the P2’ position is optimal with isoleucine, a hydrophobic amino acid. In sum, with a hydrophobic amino acid in P1’, the optimal P2 amino acid is aliphatic, and the optimal P2’ amino acid is polar; whereas a P1’ proline switches the P2 and P2’ preferences.

Surprisingly, the explanation for this switch in amino acid specificity already resided in the PDB database of protein structures, in structures of catalytically inactive HIV-1 proteases bound to a number of cleavage site peptide sequences, determined in the Schiffer lab [80]. The answer was clear. The presence of a P1’ proline switched the orientation of the side chains of the P2 and P2’ amino acids relative to that seen when P1’ was a hydrophobic amino acid (Figure 6). Since the viral protease is a dimer of two identical subunits, the binding sites in the protease that interact with the P2 and P2’ side chains (termed the S2 and S2’ subsites) are, to a first approximation, equivalent. Our analysis showed, however, that these sites must be bispecific with one face of the subsites interacting with hydrophobic/aliphatic amino acids and the opposite face of the subsites interacting with polar amino acids and/or the peptide backbone (Figure 6).

Two more points can be added to the specificity switch between the P2 and P2’ amino acid side chains that occurs as a function of the P1’ amino acid. First, we were able to take other HIV-1 protease cleavage sites and, in a predictable way, increase their cleavage rate by making substitutions to bring the sites in line with these P2/P2’ rules determined by the nature of the P1’ amino acid. Since individual cleavage site sequences are relatively highly conserved, this shows that “mismatched” amino acids in nonoptimal cleavage sites are selected for, either because of a post-cleavage role for those amino acids (as with CA Pro1) or because these suboptimal amino acids are used to slow the relative rate of cleavage at these sites. Second, by examining the P2 and P2’ amino acids of diverse retroviruses, we could infer the specificity of the protease in the S2 and S2’ subsites. The specificities described here appear to be conserved among the primate lentiviruses; however, more distant retroviruses may use another interaction strategy in which both faces of the S2/S2’ subsites exhibit preferences for hydrophobic interactions.

In closing, these two different lines of research have provided insights into protease cleavage rate determinants and the role of the processed amino-terminal proline of CA in capsid assembly and function. Both were built on the work of Stephan Oroszlan in defining the ends of retroviral proteins. Others will document the full breadth of Steve’s interests and contributions, which went well beyond what we have discussed. For our fields, the insights from Steve’s work were necessary milestones on our own paths that we are happy to acknowledge and for which we will always be grateful. There is a long history in science of naming something after the discoverer or developer, such as the Kozak sequence, Hirt extraction, Southern blot, Sanger sequencing, and many more. In this remembrance, it seems natural to assign the name of the multi-functional, conserved CA Pro1 amino acid as the Oroszlan Pro.

## Figures and Tables

**Figure 1 viruses-13-02218-f001:**
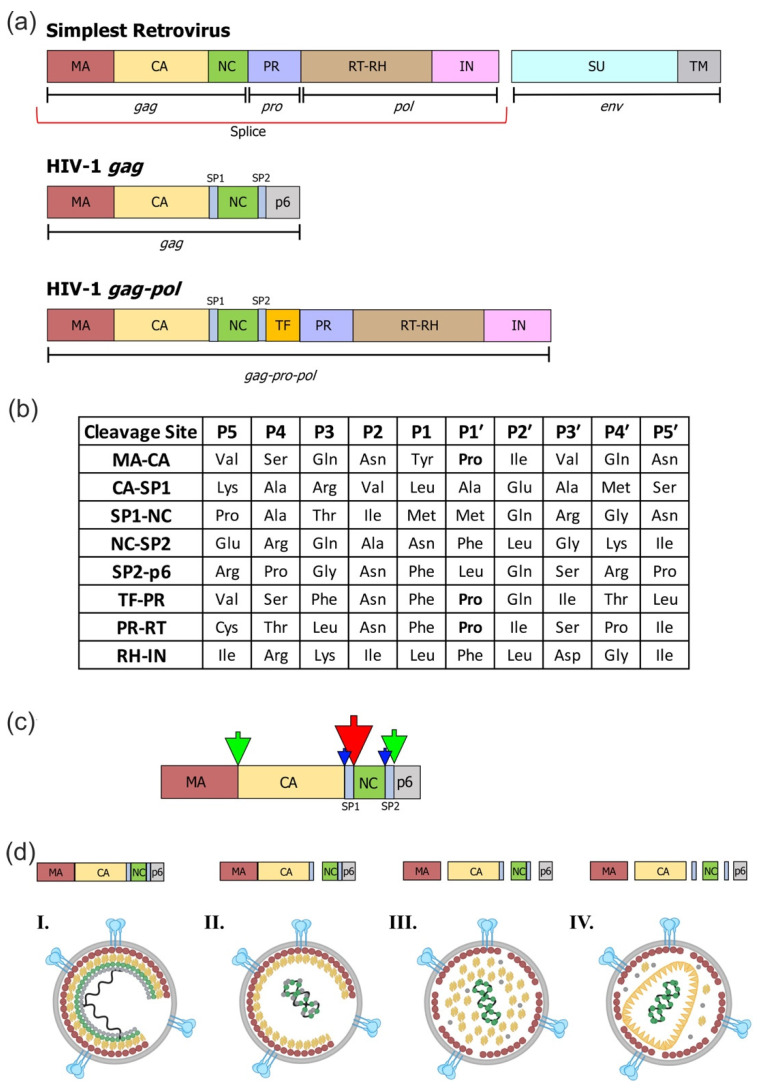
The Gag Polyprotein Precursor, Its Protease Cleavage Sites, and Proteolytic Processing During Virion Maturation. (**a**) The simplest organization of the retroviral Gag, Pro, and Pol coding domains consisting of MA (matrix), CA (capsid), NC (nucleocapsid), PR (protease), RT-RH (reverse transcriptase with its associated RNase H activity), IN (integrase), and the subunits of the Env protein SU (surface) and TM (transmembrane). The red line indicates the RNA splicing event that creates the subgenomic mRNA needed to express the *env* gene. The Gag protein and the *gag* gene have separate writing conventions to designate the protein versus the gene. The HIV-1 Gag protein has two additional spacer peptides (SP1 and SP2), as well as the “late domain” p6 polypeptide, which is involved in virion budding. The Gag-Pro-Pol precursor is formed by a frameshifting event at the NC/SP1 boundary that reads the p6 domain in an alternative reading frame (Transframe/TF), which is in frame with the *pro* and *pol* reading frames. (**b**) The major protease cleavage site sequences in the HIV-1 Gag and Gag-Pro-Pol precursors. The cleavage site sequences are written N to C, with the cleavage occurring between P1 and P1’. These cleavage site sequences are highly conserved as individual sequences even though they vary significantly between cleavage sites. The P1’ prolines are bolded. (**c**) The cleavage sites in HIV-1 Gag are shown, with the large red arrow indicating the first cleavage at the SP1/NC junction, the green arrows show the next cleavages between MA/CA and SP2/p6, and the small blue arrows show the slow removal of the two spacer peptides. The resulting processing intermediates are shown below from left to right. (**d**) Intact Gag and an immature virus particle are shown on the left, then the sequential cleavage intermediates and changes in virion protein arrangement are shown until a mature virion is formed.

**Figure 2 viruses-13-02218-f002:**
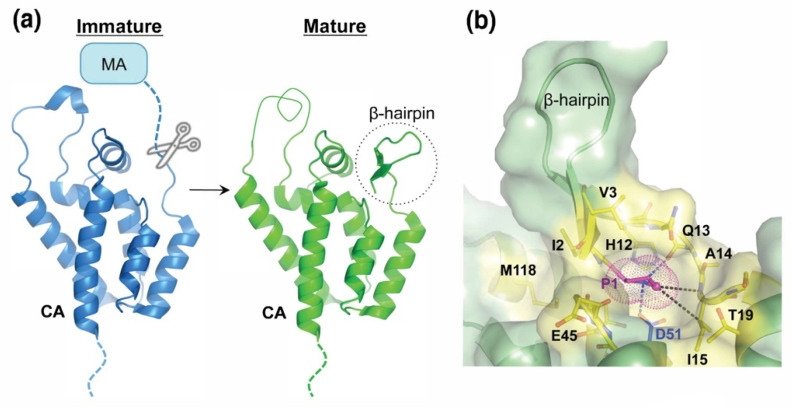
Proteolytic Processing at the HIV-1 MA-CA Junction Refolds the HIV-1 CA N-terminus into a β-hairpin Conformation that Buries the N-terminal CA Pro 1 Residue. (**a**) Proteolytic processing (scissors) of the immature Gag polyprotein at the MA–CA junction of the Gag_1-278_ polyprotein (blue, PDB: 2GOL) creates a new CA N-terminus and drives folding of the N-terminal CA β-hairpin (dashed circle, green, PDB: 2GON). (**b**) The mature N-terminal CA Pro1 (P1) fits snugly into a pocket in which the buried, protonated N-terminus makes a salt bridge with Asp51 and a hydrogen bond with Gln13 (blue dashed lines), and Pro1 Cδ makes hydrophobic packing interactions with Ala14 and Ile15 (black dashed lines, PDB: 4B4N).

**Figure 3 viruses-13-02218-f003:**
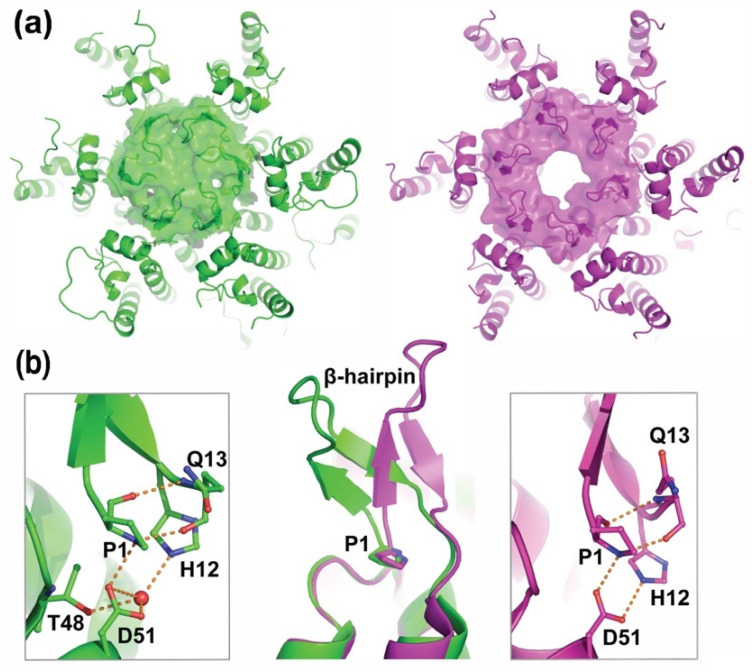
The HIV-1 CA β-hairpin Adopts Different Conformations that Open and Close the Central CA Hexamer Pore. (**a**) Space-filling model showing the central pore of the CA hexamer in its closed (green, PDB: 5TSX) and open (magenta, PDB: 5HGL) configurations. (**b**) Positions of the β-hairpin (middle panel) in the closed (green, PDB: 4B4N) and open (magenta, PDB: 5HGL) conformations. Note the rotation about CA Pro1 (P1, sticks). Expanded panels show the Pro1 polar interactions (dashed orange lines) and surrounding CA residues in the closed (green, left panel, PDB: 2GON) and open (magenta, right panel, PDB: 5HGL) conformations. The red sphere depicts a water molecule.

**Figure 4 viruses-13-02218-f004:**
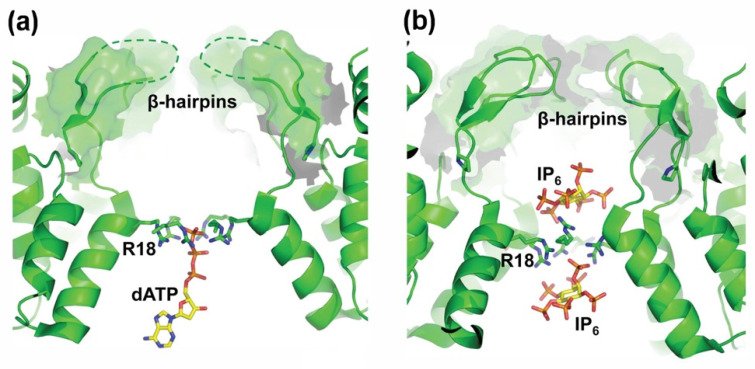
HIV-1 CA Residue R18 Forms a Basic Collar that Binds Translocating Nucleotide Triphosphates and IP6. (**a**) Cross-section of the mature CA hexamer bound to dATP, highlighting how the basic R18 “arginine collar” residues (sticks) interact with the γ- and β-phosphates of a translocating nucleotide triphosphate (PDB: 5HGM). Dashed lines depict the poorly structured loop of the β-hairpins in the closed conformation. Note that for clarity, we are showing only one of the six rotationally disordered dATP ligands. (**b**) Cross-section of the mature CA hexamer highlighting how IP6 can bind above and below the arginine collar (PDB: 6BHT). Note that for clarity, we are showing only one of each of the six rotationally disordered IP6 ligands.

**Figure 5 viruses-13-02218-f005:**
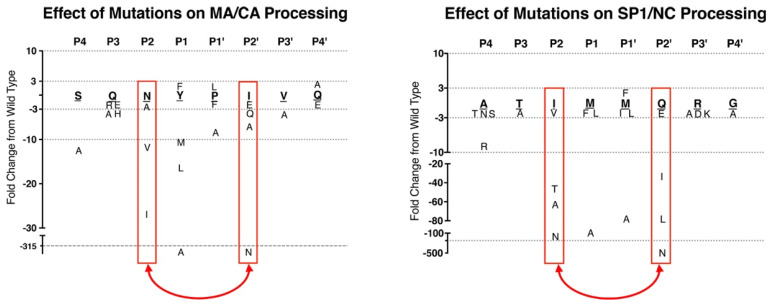
Relative Rates of Protease Cleavage of SP1/NC and MA/CA Cleavage Sites with Substitutions. The wild-type sequence is shown in bold and underlined (MA/CA: **SQNY/PIVQ**; SP1/NC: **ATIM/MQRG**). The substitutions are shown above or below in single-letter amino acid code. The change in the rate of cleavage relative to wild type is shown by the vertical positioning of the indicated substitution, with faster rates above the wild-type sequence and slower rates below, as indicated by the *Y*-axis label. Changes within three-fold represent little to no change. The P2 and P2’ positions are indicated by the red rectangles. The red double-headed arrow notes the change in specificity in P2 and P2’. Data are reproduced from ref [79].

**Figure 6 viruses-13-02218-f006:**
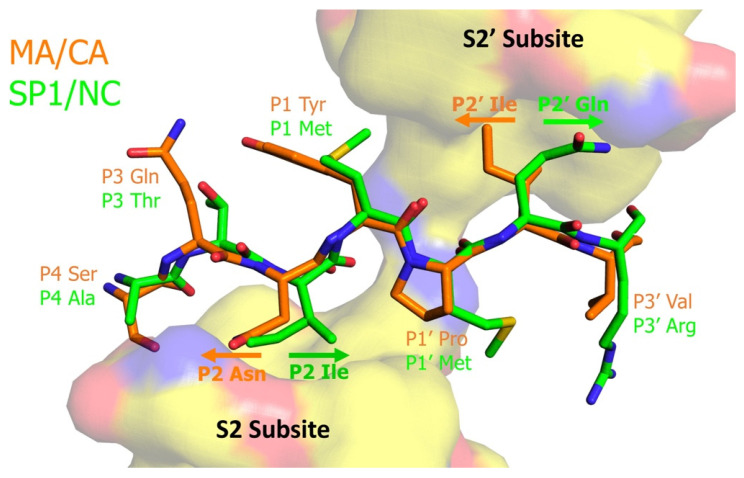
Structural Interpretation of Substrate Orientation Within the HIV-1 Protease with and without a P1’ Proline. The surfaces of the viral protease are shown for the S2 and the S2’ subsites with yellow as hydrophobic, blue as positive charge, and red as negative charge. Two substrates are superimposed in stick figure with MA/CA in orange and SP1/NC in green. The individual substrate amino acids are indicated, with arrows showing the differing orientations of the P2 and P2’ side chains in the substrate. The structures are from ref. [80], and the figure was constructed in PyMol using PDB files 1KJ4 and 1KJ7.

**Table 1 viruses-13-02218-t001:** Retroviral Protease Cleavage Sites Upstream of CA ^a^.

BLV	MA/CA	PAIL/PIIS
EIAV	MA/CA	SEEY/PIMI
HTLV-I	MA/CA	PQVL/PVMH
HTLV-II	MA/CA	TQCF/PILH
HIV-1	MA/CA	SQNY/PIVQ
MMTV	n/CA	TFTF/PVVF
MPMV	p12/CA	KDIF/PVTE
MuLV	p12/CA	SQAF/PLRA
RSV	p10/CA	VVAM/PVVI
SIVmac	MA/CA	GGNY/PVQQ

^a^ Summarized from reference [43] and references therein.

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
