# Peer review of "Stephan Oroszlan and the Proteolytic Processing of Retroviral Proteins: Following A Pro"

_viruses, 2021, doi:10.3390/v13112218_

Round 1

Reviewer 1 Report

In this remembrance, the authors outline some of the contributions made to the retrovirology field by the late Steve Oroszlan, as an example related to the identification of proline as the first residue of retroviral capsid (CA) proteins.  The article was a pleasure to read and graciously describes some of the contributions of this outstanding scientist.  It will be a valuable contribution to the special issue of Viruses devoted to Steve.  I have only a few very minor comments.

  1. Line 171, the authors refer to the betulinic acid compounds (e.g., bevirimat) as capsid inhibitors. This may confuse readers who will be more accustomed to their being referred to as maturation inhibitors, in contrast to compounds like PF74 or GS-6207 that target capsid (described in detail by Kleinpeter et al. Viruses 2020).
  2. Line 180. Perhaps replace “fresh” CA N-terminus with something like “newly generated” CA N-terminus?
  3. Line 206 “well defined hydrophobic within”. Word missing, “pocket” or “groove”.

Author Response

The three small changes were made as suggested

Reviewer 2 Report

The article, written by two distinguished scientists, outlining the importance of Steve Oroszlan’s pioneering work influenced the field and their own research, is welcome at two levels. First, all too often our junior colleagues (and sometimes their more senior mentors) do not understand the foundations of our current knowledge. Pointing out how important it was for Steve and his colleagues to accurately define the ends of the structural proteins and the viral enzymes is entirely appropriate. Second, it is also useful to link that knowledge to an up-to-date view of how the various steps in proteolysis are regulated and how that affects the conversion of the immature and mature virion. My only suggestions are intended to clarify things that the authors know well but have presented in a way that could confuse a naïve reader.

  • Figure 1, Panel a. Neither the figure legend nor the text explains how Gag to Gag-Pol are made. Adding a sentence or two would help, and/or the panel could be modified to show both Gag and Gag-Pol. More confusing to someone who doesn’t understand the way retroviral polyproteins are made is the apparent linkage, in the figure, of IN and Su. Gag-Pol should not be linked to Env in the drawing.
  • Figure 1, Panel d. Remove “Lorem Ipsum” from the figure.
  • Figure 1 panel d. Although the RNA dimer is stabilized during virion maturation, the initial dimerization of the RNA takes place before the immature virion is produced.
  • Figure 5, both panels. It would help to have the amino acid sequences of the substrates given at the top of each panel. The sequence data are in figure 1 but going back and forth between the figures is not optimal for the reader.
  • Although it is a very minor point, although Steve Oroszlan’s lab was at the NCI-Frederick facility, he never worked directly for the NCI (page 2, line 88).
  • Putting the legend to Figure 4 on two pages is not optimal.

Author Response

We have tried to improve Figure 1, the legend, and the text to clarify how Gag and Gag-Pro-Pol fit into the virus life cycle.

Figure 5 already has the wild type sequence in bold. However, this may be too subtle since the reviewer missed it. We have added underlining and will include the sites in the legend.

We have removed the reference to NCI since the details of Dr. Oroszlan's affiliations are covered in other contributions.

Reviewer 3 Report

The review article by Swanstrom and Sundquist is an excellent tribute to Steve Oroszlan’s work in the area of retrovirus proteolytic processing and maturation, befitting his scientific legacy. I enjoyed reading it and will certainly recommend it to trainees as an introduction to HIV-1 maturation. I myself will shamelessly pilfer Figure 1 for use in lectures.

I offer the following minor suggestions:

  1. Typo in line 99: “nuculeocapsid”
  2. Starting in line 327 with “Recently, we took a more global, brute force…” I was unsure whether the authors were describing new or previously published data. Adding the citation to the end of that sentence could make that clear.
  3. “Oroszlan Pro” doesn’t quite roll off the tongue. Therefore, I will probably refer to the CA N-terminal proline as the “O-Pro.”  

Author Response

We corrected the first two points.

The reviewer is welcome to make this change in their reference to the N terminus of CA but I suspect the naming of that amino acid will have a life in this retrospective but probably not far beyond.

Reviewer 4 Report

This a highly focused review of the implications for one of Steve Oroszlan's discoveries, the conserved N-terminal Pro of the retroviral CA protein.  The review is a complicated mix of overview and perspective on the one hand, and biochemical or structural detail on the other. Overall, what the authors have come up with is a very nice tribute to this pioneer biochemist in biochemical retrovirology.  The introductory paragraph is an example of the fine writing that is evident throughout. I have only very minor suggestions that the authors should consider. Overall, chapeaux bas to the authors.

--Line76: in the phrase "most typically based on gene sequences", it would be good add something to clarify that this refers to the present and recent past, but of course not to the time when Oroszlan was determining N-terminal sequences by classical chemistry (Edman degradation).

--Line99 figure legend: spelling of nucleocapsid

--Line106: In my opinion the sentence "these cleavage site sequences are highly conserved" should be removed. Even a casual naive reader who glances at the table can see that the sites are NOT conserved. And then the entire paragraph starting at line 127 is devoted to explaining that.

--line140:  insert "that" after "evidence".

--Line230: the verb should be "explains", since "none" is singular, meaning "no one".

--Line397: "Since cleavage site sequences..."  Again, in my opinion that introductory clause is not needed or wanted.

Author Response

All things changed as indicated except for one point raised twice.

We have clarified what we mean in that the individual cleavage sites are highly conserved even if they are very different between sites.  This is an important concept because many of the sites are suboptimal but their conservation suggested this is selected for.  MA/CA is the extreme case in the need for the P1' Pro.  This site could be made to cleave faster except for that requirement.  The suboptimal rates of cleavage at other sites suggest they too are under selective pressure of unknown nature to be slower than they could be.  We have tried to clarify the within site conservation as opposed to the between site variation.